# Ensuring privacy and confidentiality of cloud data: A comparative analysis of diverse cryptographic solutions based on run time trend

**John Kwao Dawson** [1,2] *, **Twum Frimpong**[2], **James Benjamin Hayfron Acquah**[2], **Yaw Marfo Missah**[2]

**1** Sunyani Technical University, Sunyani, Ghana, **2** Kwame Nkrumah University of Science and Technology, Kumasi, Ghana

* kwaodawson@stu.edu.gh

**Data Availability Statement:** https://www.kaggle.com/datasets/morriswongch/kaggle-datasets.

**Funding:** The authors received no specific funding for this work.

## Abstract

The cloud is becoming a hub for sensitive data as technology develops, making it increasingly vulnerable, especially as more people get access. Data should be protected and secured since a larger number of individuals utilize the cloud for a variety of purposes. Confidentiality and privacy of data is attained through the use of cryptographic techniques. While each cryptographic method completes the same objective, they all employ different amounts of CPU, memory, throughput, encryption, and decryption times. It is necessary to contrast the various possibilities in order to choose the optimal cryptographic algorithm. An integrated data size of $5n*10^2$ ($KB$ ($\in$ 1,2,4,10,20,40) is evaluated in this article. Performance metrics including run time, memory use, and throughput time were used in the comparison. To determine the effectiveness of each cryptographic technique, the data sizes were run fifteen (15) times, and the mean simulation results were then reported. In terms of run time trend, NCS is superior to the other algorithms according to Friedman's test and Bonferroni's Post Hoc test.

## 1.0 Introduction

Human activity has risen, making communication more difficult, and necessitating data protection. [1]. A paradigm shift in data storage needs to be implemented in order to secure these enormous amount of data [2]. Due to the enormous amount of data produced by numerous social media platforms, including Facebook, Twitter, Instagram, and e-commerce websites, cloud computing is currently the preferred option [3].

Amazon's four-hour cloud computing downtime in 2017 cost S&P 500 Company $150 million, according to a Maeser [4]. A network traffic control organization called Apica predicted that the top 54 e-commerce sites will experience a decline in activity of at least 20% [4]. According to Ponemon, Fortune 1000 companies lost just over $ 2.5 billion in 2015 as a result of data center shutdowns brought on by hackers. According to Maeser [4], the need for cloud computing will increase by about 266% between 2013 and 2020 as a result of the massive volumes of data that the Internet of Things will produce.

**Competing interests:** The authors have declared that no competing interests exist.

Once more, Maeser [4] stressed that the infrastructure-as-a-service aspect of cloud computing will result in an increase in demand of roughly 85%.

Because of the benefits of agility, scalability, availability, accelerating the development of work, and lowering operating costs by utilizing pay-as-you-use services, cloud computing continues to gain popularity over traditional on-site data centers [5, 6]. As a result, Information Technology giants are now investing far more money on cloud computing than they did in the past. These advantages have led businesses to use cloud services such as Software-as-a-Service (SaaS), Infrastructure-as-a-Service (IaaS), Platform-as-a-Service, and Container-as-a-Service (CaaS) for their pay-as-you-use activities on the cloud [7–9].

The adoption of cloud computing is accompanied by a number of security issues, such as data privacy and confidentiality [10–12]. In order to guarantee the secrecy and privacy of cloud data, cryptographic techniques have shown to be an effective and efficient methods [12–16, 18].

In this study, the symmetric algorithms Enhanced RSA (ERSA) [17], Non-Deterministic Cryptographic Scheme (NCS) [18], Enhanced Homomorphic Scheme (EHS) [19], Chacaha20, and Salsa20 are compared based on run time trend, throughput time, and memory usage to determine which algorithm among them could be used to ensure the confidentiality and privacy of cloud data.

## 1.1 Identified problem

The biggest problem with cloud computing has been data security. Researchers have suggested many encryption technique variations to protect cloud data [20]. Enhanced RSA [17], Non-Deterministic Cryptographic Scheme [18], Enhanced Homomorphism Scheme [19], Chacha20, and Salsa20 are a few examples of these techniques. These suggested encryption techniques are effective in limiting unauthorized access to sensitive information.

In terms of run time trend, throughput time, and memory complexity, it is unclear which one performs better than the other. Such knowledge is essential for industry startups, other professionals, and researchers who are interested in utilizing effective algorithms to protect privacy and confidentiality of data in the cloud. Therefore, the major goal of this research is to test Enhanced RSA, Non-Deterministic Cryptographic Schemes, Enhanced Homomorphism Schemes, Chacha20, and Salsa20 in order to determine the computational statistics of the best method. Once more, this study offers a solid framework that theoretically and practically combines all of the recognized algorithms into a robust system. The principal contribution of this paper is the proposition of a comprehensive cryptographic scheme(s) that can be used to ensure confidentiality and privacy of cloud data.

## 2.0 Literature review

Cloud computing users save their data in the cloud making it a remote location based system. It compromises secrecy, which is essential for cloud computing to be acceptable. To boost security and trustworthiness, cryptography is often used in cloud computing.

ALmarwani et al. [21] presented a unique tagging approach called Tagging of Outsourced Data (TOD) in an endeavor to protect the secrecy of data stored in the cloud. Their method supported cloud data through verification. Their method had a short run time, enabling for widespread use by mobile devices. Tahir et al. [22] presented a genetic algorithm called CryptoGa to help Almarwani et al. achieve data privacy. When compared to state-of-the-art algorithms like AES, RSA, and DES, their approach had shorter execution times.

Shen et al. [23] advocated using proxy re-encryption and Oblivious Random Access Memory (ORAM). Their technique was designed to ensure multi-user data sharing on the cloud.

The ciphertext gained through the proxy re-encryption enabled members to regulate access and, as a consequence, establish data privacy.

Garad et al. [24] suggested a cryptosystem to protect submitted files to the cloud server. AES-CCM, AES-GCM, and CHACHA20_POLY1305 were the asymmetric cryptographic algorithms they employed. They divide the file into N pieces, then use various cryptographic techniques to encrypt each portion. Thabit, et al. [25], ensured the confidentiality and privacy of cloud data by proposing a Lightweight Cryptographic Algorithm. Their algorithm integrated Feistel and substitution schemes to raise the encryption complexity of their algorithm. Their algorithm was very effective regarding run time. Tiwari and Neogi [26] proposed a security scheme that secured a multi-tenant hybrid cloud by combining Kerberos Authentication Protocol with Resource Allocation Manager Unit (RAMU). Their scheme allowed for more resource access while also improving client confidentiality and security. The model validates the user's request before providing access, preventing the password from being revealed to hackers during transmission. The Key Distribution Centre (KDC) validates the request and RAMU grants access after reviewing the control database and resource allocation map. Gadde et al. [27] suggested an Improved Blowfish cryptography strategy for encrypting and decrypting sensitive data in the cloud server using the optimum key. Optimal key creation is a critical method for achieving integrity and confidentiality goals of integrity and confidentiality. Similarly, data restoration is the inverse process of sanitization (decryption). In [28] a four-step data security approach in cloud computing was proposed. They used the least significant bit of LSB approaches to integrate three cryptographic algorithms, RSA, AES, and identity-based encryption, with steganography to attain confidentiality and privacy of cloud data.

## 3.0 Methodology

### 3.1 Enhanced RSA

Enhanced RSA Improved traditional RSA's security by combining classic RSA with Gaussian interpolation formula. The integration raises the security of RSA to the fifth level. After encrypting the message's ASCII values with Gaussian First Forward interpolation, the conventional RSA is used to encrypt and decode the message at the second and third levels. The last stage uses Gaussian First Backward interpolation to decode the data again, as seen in Fig 1. The integration helps to overcome the classic RSA factorization problem [17].

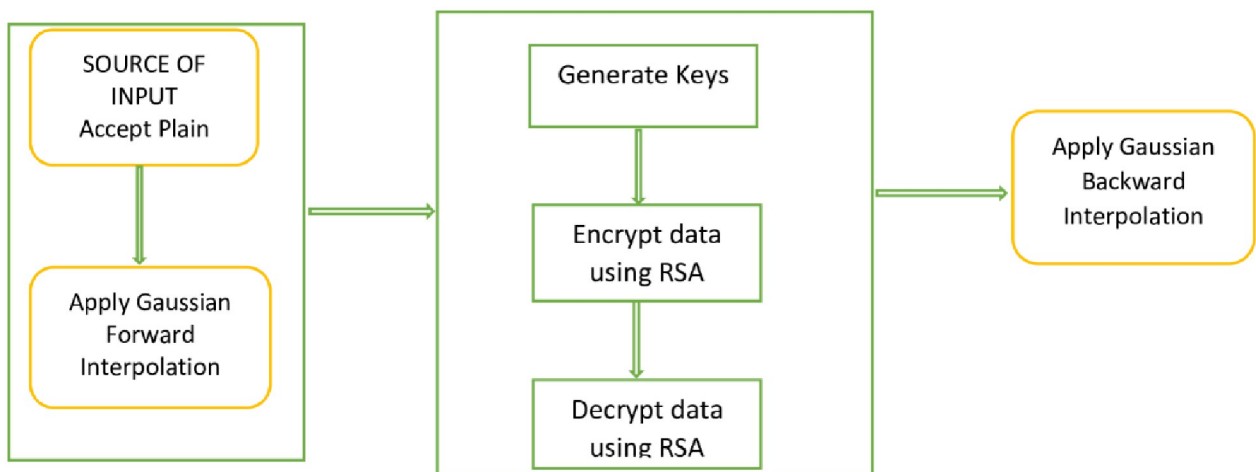

**Fig 1. Work flow diagram of enhanced RSA [17].**

## 3.2 Non-Deterministic Cryptographic Scheme

This method consists of three stages: key generation, encryption, and decryption. The three levels of key production are meant to produce secret keys that help to secure the algorithm [18]. These include the use of Good Prime numbers, the Linear Congruential Generator, the Fixed Sliding Window Algorithm, and XORing the output with the plaintext. In NCS a sub array of $\left(\frac{n(a[i])}{4}\right)$ is computed on the twelve numbers generated after the application of the Fixed Sliding Window [18].

## 3.3 Enhanced Homomorphism Scheme

Enhanced Homomorphism Scheme (EHS) is developed and implemented by the amalgamation of Good Prime Numbers (GPN), Linear Congruential Generator (LCG), Fixed Sliding Window Algorithm (FSWA), and Gentry's Algorithm. Two stages are considered in this algorithm, the generation of keys and the application of the homomorphism scheme. Three procedures are used to generate the keys which includes the generation of two good prime numbers with the product as a seed for the Linear Congruential Generator to produce twelve numbers. The sliding window algorithm is applied on the twelve numbers using a sub-array of three $\left(\frac{n(a[i])}{3}\right)$. The first value is $s_i$, second value $s_j$, the third value $s_k$, fourth value is $s_l$ and with M the plaintext as seen in Eq 1 for data encryption [19].

$$C = M + s_i * s_j + s_k * s_l \tag{1}$$

## 3.4 Salsa20

The Salsa20 stream cipher, initially developed by Bernstein in 2008, has 20 rounds, while it's more recent variations, Salsa20/8 and Salsa20/12, have 8 and 12 rounds respectively. Salsa20/20 refers to the 20-round Salsa20. The Salsa20 stream cipher accepts keys of 128 and 256 bits. The Salsa20 core is composed of a 256-bit key (n0, n1, n2, n3, n4, n5, n6, n7), a 64-bit block counter (e0, e1), a 64-bit nonce (d0, d1), and four 32-bit diagonal constants (c0, c1, c2, c3) that may be mapped into a 4 4 matrix as in Eq 2.

$$X = \begin{pmatrix} b_0^0 & b_1^0 & b_2^0 & b_3^0 \\ b_4^0 & b_5^0 & b_6^0 & b_7^0 \\ b_8^0 & b_9^0 & b_{10}^0 & b_{11}^0 \\ b_{12}^0 & b_{13}^0 & b_{14}^0 & b_{15}^0 \end{pmatrix} \rightarrow \begin{pmatrix} c_0 & n_0 & n_1 & n_2 \\ n_3 & c_1 & d_0 & d_1 \\ e_0 & e_1 & c_2 & n_4 \\ n_5 & n_6 & n_7 & c_3 \end{pmatrix} \tag{2}$$

The three operations of Salsa20, addition, rotation, and XOR, give desired cryptographic features, with modular addition providing non-linearity and bit rotation providing diffusion in a word. The diffusion attribute is propagated from one word to the next via the XOR technique [29].

## 3.5 Chacha20

Chacha20 is a 2008 update of Salsa20 that uses a new round function to boost diffusion. Salsa20 uses a 32-bit module addition, XOR, and rotation operations based core hash function, whereas Chacha20 uses an internal state of sixteen 32-bit words arranged as a $4 \times 4$ matrix to map a 256-bit key (128-bit key is also suitable for Salsa20), a 64-bit nonce, and a 64-bit counter to a 512-bit block key stream. While the configurations differ, their beginning states are both made up of 8 words of key, 2 words of stream position (counter), 2 words of nonce, and 4 words of constant. Chacha20 is a replacement for Salsa20's quarterround, which updates each word twice with the same amount of operations as shown in Eq 3. Unlike Salsa20, which

alternates quarterrounds down columns and across rows, Chacha20 executes quarterrounds down columns and along diagonals in a doubleround. Chacha20 likewise employs 10 double-round iterations and the same output function [30].

$$\left.\begin{cases} k+=x; & z\oplus=k; & z\lll=16; \\ y+=z; & x\oplus=y; & x\lll=12; \\ k+=x; & z\oplus=k; & z\lll=8; \\ y+=z; & x\oplus=y; & x\lll=7; \end{cases}\right\} \quad (3)$$

### 3.6 The proposed framework of the system

This section provides a broad summary of the comparative study of the enhanced RSA, Non-Deterministic Cryptographic Scheme, enhanced Homomorphism Scheme, Chacha20, and Salsa20. For the ERSA, NCS, EHS, ChaCa20, and Salsa20, the architecture is divided into five phases: key generation, encryption, decryption, memory utilization, and throughput. From Fig 2, the user registers with a cloud service provider. The registered client uploads the plaintext onto the cloud after encryption using either ERSA, NCS, EHS, Chaca20, or Salsa20 to obtain Ciphertext from which Encryption time, memory usage, and throughput time are computed. This is achieved using any digital device [6]. The decryption time, memory storage

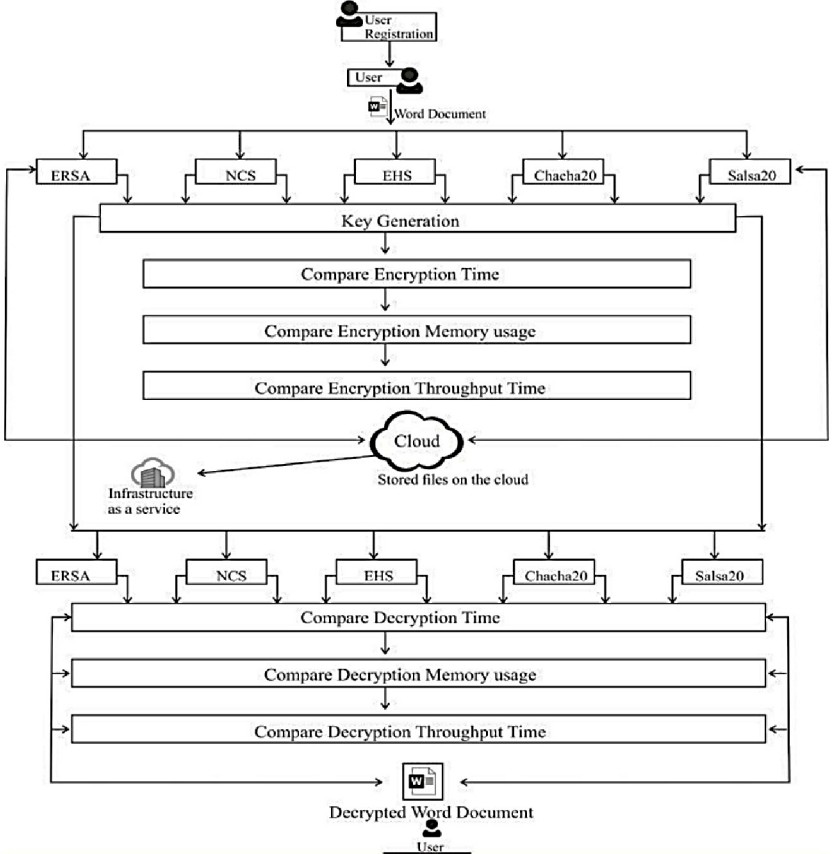

**Fig 2. Framework for the proposed model.**

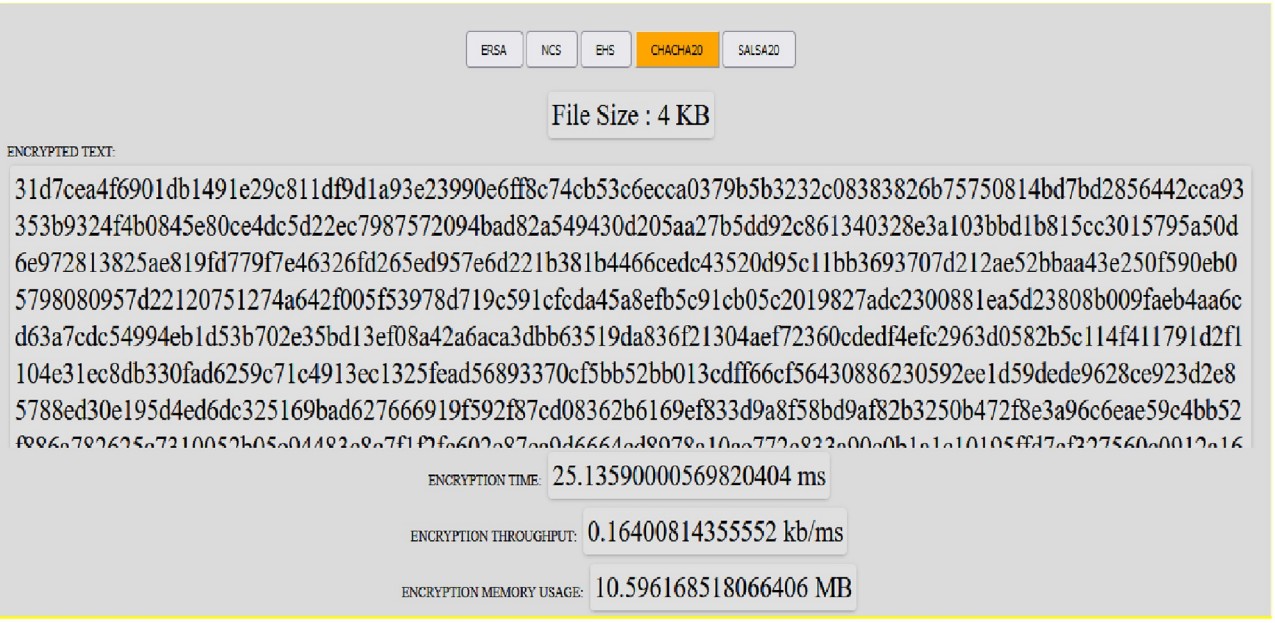

**Fig 3. Encrypted performance for Chacha20 using 4KB data.**

time, and throughput time are calculated once the Ciphertext has been downloaded. Figs 3 and 4 show a snapshot of the run time for 4KB data for Chacha20 using the system, with further information available at https://github.com/Elkie1/Chacha20.

## 4.0 Experimentation

The comparative analysis of the Enhanced RSA [17], Non-Deterministic Cryptographic Scheme [18], Enhanced Homomorphism Scheme [19], Salsa20 and Chacha20 was implemented on an i7 Lenovo computer, 2.10 GHz CPU using C# language. C# programming

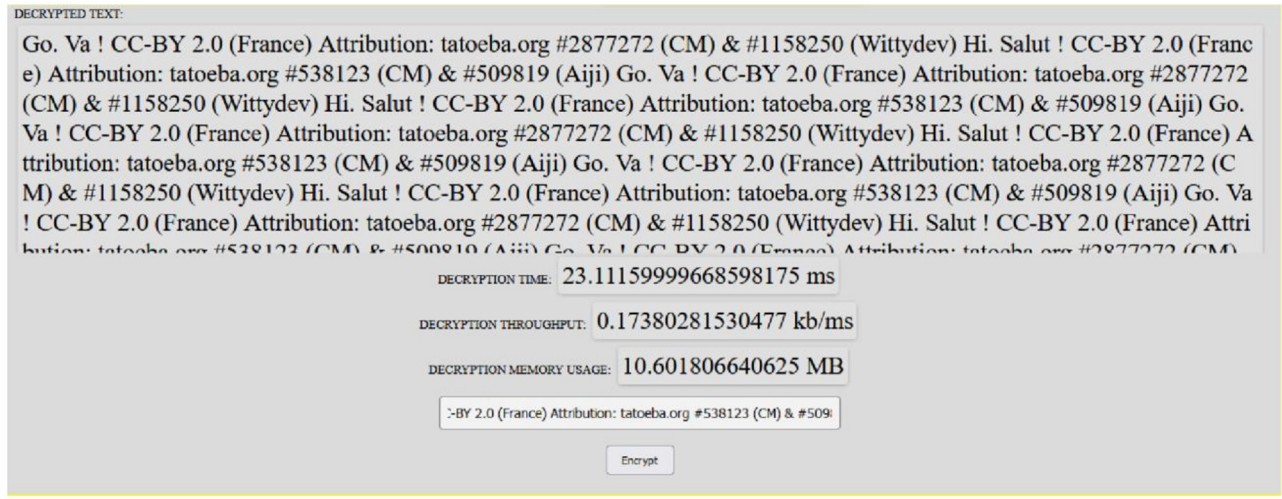

**Fig 4. Decrypted performance for Chacha20 using 4KB data.**

language is preferable because it influences the execution time of data as was implemented in a NET C# programming language where it was used to test the execution of AES algorithm resulting in 300MB/seconds while OpenSSL C simulation produced a 960 MB/seconds average speed [31].

## 4.1 Description of dataset used in this work

The study's dataset was collected from the Kaggle database [32]. The dataset was used to assess the robustness of the algorithms in terms of run time trend, memory use, and throughput. The dataset is an English-to-French translation that includes text, numbers, and special characters. This is critical since Loyka et al.'s [33] investigation showed divergent results when only text and numbers were utilized. The proposed algorithms were evaluated using data sizes of $5n*10^2$ ($KB (\in 1,2,4,10,20,40)$. The dataset was run fifteen (15) times to ensure the accuracy of the run time, and the mean and standard deviation of the execution were computed.

## 5.0 Results

### 5.1 Encryption time

From Table 1, Chacha20 had the lowest mean encryption time of 15.9047±1.69 milliseconds followed by NCS (52.13±31.1766) with Salsa20 having the highest mean encryption time of 853±85.06 milliseconds when data size of 500KB (0.5MB) was executed. The encryption time increased from 853±85.06 milliseconds to 1302.8 ±703.97 milliseconds for Salsa20 when the data size was increased to 1000KB making it linear [34, 35]. The encryption time for ERSA also increased from 462.93±40.93 milliseconds to 575.67±57.05 millisecond when data size was raised to 1000KB. However, the encryption time for NCS reduced from 147.33±172.41 milliseconds to 85.8±54.46 milliseconds and to 82.2±75.17 milliseconds when data size was increased from 500KB to 1000KB, and from 2000KB to 5000KB.

### 5.2 Decryption time

Table 2, presents the comparison of the mean decryption time trend for Salsa20, Chacha20, ERSA, NCS, and EHS. With data size of 500KB, NCS had the lowest decryption time of 74 ±45.16 milliseconds followed by Chacha20 (281.33±35.42) and EHS with a decryption time of 368.4±133.88 milliseconds. When the data size was increased to 1000KB, NCS had the lowest decryption time of 105.6±71.78 milliseconds with Salsa20 having the highest run time of 869.4 ±223.18. From Table 2, it could be seen that for Salsa20, Chacha20 and ERSA their run time increases as data sizes increases making them linear (O (N)). However, the run time for NCS and EHS alternates as the sizes of the data increases making it non-linear.

**Table 1. Comparing the mean encryption time for Salsa20, Chacha20, ERSA, NCS, and EHS.**

| Data Size(KB) | Salsa20 (ms) | Chacha20(ms) | ERSA (ms) [17] | NCS (ms) [18] | EHS (ms) [19] |
|---|---|---|---|---|---|
| 500 | 853±85.06 | 15.9047±1.69 | 462.93±40.93 | 52.13±31.1766 | 178.27±64.57 |
| 1000 | 1302.8 ±703.97 | 19.8653±1.65 | 575.67±57.05 | 147.33±172.41 | 204.33±132.99 |
| 2000 | 2936.2±777.81 | 195.5867±32.21 | 695.93±71.59 | 85.8±54.46 | 268.93±158.19 |
| 5000 | 5916.33±764.99 | 4186.7373±1343.42 | 738±150.31 | 82.2±75.17 | 139.27±75.45 |
| 10000 | 11536.27±2860.06 | 5995.722±1932.96 | 866.67±84.74 | 98.07±121.57 | 118.67±39.85 |
| 20000 | 32531.07±46652.36 | 7769.3793±1169.29 | 1679.4±2514.36 | 163.07±168.07 | 148.53±95.19 |

**Table 2. Comparing the mean and standard deviation decryption time for Salsa20, Chacha20, ERSA, NCS, and EHS.**

| Data Size (KB) | Salsa20 (ms) | Chacha20(ms) | ERSA (ms) [17] | NCS (ms) [18] | EHS (ms) [19] |
|---|---|---|---|---|---|
| 500 | 869.4±223.18 | 281.33±35.42 | 391.4±47.08 | 74±45.16 | 368.4±133.88 |
| 1000 | 2152.13±2469.76 | 508±73.48 | 514.2±75.87 | 105.6±71.78 | 277.4±144.83 |
| 2000 | 3020.41±1369.13 | 1524.93±2466.48 | 612.4±89.93 | 159.53±108.66 | 234.8±105.27 |
| 5000 | 5888.8±864.34 | 6138.27±1792.07 | 732.93±94.29 | 151.07±155.33 | 253.13±168.95 |
| 10000 | 10582.13±1274.53 | 10028.53±2757.07 | 1375.47±121.33 | 115.93±123.58 | 315.53±122.98 |
| 20000 | 19699.33±1068.30 | 15893.53±1934.37 | 1874.73±3528.41 | 158.87±144.66 | 137.87±71.83 |

## 5.3 Encryption throughput (KB/ms)

The number of units of data that can be processed at a given time is considered throughput [36]. This is computed using Eq 4.

$$Throughput = \frac{Size\ of\ Data}{Run\ Time} \tag{4}$$

From Table 3, with 500KB of data, Salsa20 had the lowest mean encryption throughput time of 0.039078 KB/ms with Chacha20 having the highest throughput time of 31.43731 KB/ms. Also, when the data size was increased to 2000KB, NCS had the highest mean encryption throughput time of 23.31002 KB/ms with Salsa20 having the lowest mean encryption throughput time of 0.04541KB/ms followed by ERSA with a mean encryption throughput time of 2.87383 KB/ms. However, when data size was increased to 10000KB, NCS had the highest encryption throughput 101.9714 KB/ms. However, with a data size of 20000KB, EHS had the highest mean encryption throughput time of 134.6499102 KB/ms followed by NCS with a mean encryption throughput time of 122.6492 KB/ms.

## 5.4 Decryption throughput

From Table 4, NCS had the highest mean decryption throughput time of 6.756757 KB/ms with Salsa20 having the lowest mean decryption throughput time of 0.575109 KB/ms when data size of 500KB was executed. Again when the data size was increased to 2000KB, NCS had the highest mean decryption throughput of 12.53656 KB/ms followed by EHS with a mean decryption throughput time of 8.517887564 KB/ms. However, with a data size of 20000KB, EHS had the highest mean decryption throughput time of 145.0676983 KB/ms followed by NCS with a mean decryption throughput time of 125.8917 KB/ms.

**Table 3. Comparing the mean encryption throughput (KB/ms) for Salsa20, Chacha20, ERSA, NCS, and EHS.**

| Data Size (KB) | Salsa20(KB/ms) | Chacha20(KB/ms) | ERSA (KB/ms) [17] | NCS (KB/ms) [18] | EHS (KB/ms) [19] |
|---|---|---|---|---|---|
| 500 | 0.039078 | 31.43731 | 1.080069 | 9.590793 | 2.804786836 |
| 1000 | 0.051172 | 50.33895 | 1.737116 | 6.78733 | 4.893964111 |
| 2000 | 0.04541 | 10.22565 | 2.873838 | 23.31002 | 7.436787308 |
| 5000 | 0.056341 | 1.194247 | 6.775068 | 60.82725 | 35.90234562 |
| 10000 | 0.057789 | 1.667856 | 11.53846 | 101.9714 | 84.26966292 |
| 20000 | 0.040986 | 2.574208 | 11.90902 | 122.6492 | 134.6499102 |

**Table 4. Comparing the mean decryption throughput (KB/ms) for Salsa20, Chacha20, ERSA, NCS, and EHS.**

| Data Size (KB) | Salsa20(KB/ms) | Chacha20(KB/ms) | ERSA (KB/ms) [17] | NCS (KB/ms) [18] | EHS(KB/ms)[19] |
|---|---|---|---|---|---|
| 500 | 0.575109 | 1.777251 | 1.277466 | 6.756757 | 1.357220413 |
| 1000 | 0.464655 | 1.968504 | 1.944769 | 9.469697 | 3.604902668 |
| 2000 | 0.662161 | 1.311533 | 3.265839 | 12.53656 | 8.517887564 |
| 5000 | 0.849069 | 0.814562 | 6.821903 | 33.09797 | 19.75243613 |
| 10000 | 0.9449 | 0.997155 | 7.27062 | 86.25647 | 31.6923727 |
| 20000 | 1.015263 | 1.258373 | 10.66818 | 125.8917 | 145.0676983 |

## 5.5 Encryption memory usage

From Table 5, with a data size of 500KB, Salsa20 has high memory complexity of 196.4±32.24 megabytes followed by Chcaha20 (191.8±35.91 megabytes) with EHS having the lowest memory complexity of 16±2.75 megabytes. When data size was increased to 20000KB, Salsa20 used 6160.27±650.17 megabytes of memory which was still the highest followed by Chacha20 (6092.27±653.58) while NCS used the least memory complexity of 16.4±3.78 megabytes.

## 5.6 Decryption memory usage

From Table 6, Salsa20 has the highest mean memory complexity of 192.13±34.41 megabytes of memory when 500KB of data was decrypted with NCS having the lowest mean memory complexity of 15.87±3.04 megabytes. However, when data size was increased to 20000 KB, Chacha20 had the highest memory complexity of 6281.4±713.08 megabytes with NCS still having the lowest memory complexity of 17.07±3.28 megabytes.

## 5.7 Comparing significance difference between the encryption and decryption times using Friedman Test and Bonferroni Post Hoc test

The Friedman's Test tests the hypothesis that "all treatment effects are zero" as against the alternate hypothesis "not all treatment effects are zero". From the output in Tables 7 and 9,

**Table 5. Comparing the mean encryption memory usage for Salsa20, Chacha20, ERSA, NCS, and EHS (MB).**

| Data Size (KB) | Salsa20 (MB) | Chacha20(MB) | ERSA (MB) [17] | NCS (MB) [18] | EHS (MB) [19] |
|---|---|---|---|---|---|
| 500 | 196.4±32.24 | 191.8±35.91 | 20.6±2.59 | 17.13±1.99 | 16±2.75 |
| 1000 | 270.33±34.78 | 265.13±38.54 | 20.6±2.47 | 16.13±3.87 | 16.13±3.64 |
| 2000 | 588.6±66.89 | 597.67±66.57 | 20.6±2.13 | 16.33±3.75 | 16.53±3.25 |
| 5000 | 1443.2±192.69 | 1392.33±224.02 | 20.13±2.07 | 16.13±3.98 | 16.47±3.62 |
| 10000 | 3436.13±355.35 | 3453.13±365.96 | 21.33±2.09 | 15.87±3.44 | 17.4±3.62 |
| 20000 | 6160.27±650.17 | 6092.27±653.58 | 20.73±2.46 | 16.4±3.78 | 17.6±3.12 |

**Table 6. Comparing the mean decryption memory usage for Salsa20, Chacha20, ERSA, NCS, and EHS (MB).**

| Data Size (KB) | Salsa20(MB) | Chacha20(MB) | ERSA (MB) [17] | NCS (MB) [18] | EHS (MB) [19] |
|---|---|---|---|---|---|
| 500 | 192.13±34.41 | 191.87±35.89 | 20.67±2.16 | 15.87±3.04 | 16.2±3.05 |
| 1000 | 270±34.78 | 259.47±40.63 | 20.23±2.37 | 16.13±3.87 | 17.5±3.74 |
| 2000 | 595.13±68.94 | 587.47±78.23 | 20.6±2.32 | 16.47±3.6 | 17.6±3.15 |
| 5000 | 1360.53±236.62 | 1384.53±241.7 | 20.87±2.13 | 16.33±3.79 | 17.6±3.64 |
| 10000 | 3453.73±368.56 | 3404.87±374.45 | 20.53±2.19 | 16.2±3.14 | 17.7±3.13 |
| 20000 | 6267.13±732.39 | 6281.4±713.08 | 21.2±2.83 | 17.07±3.28 | 18.2±2.68 |

**Table 7. Friedman test of the encryption times for Chacha20, Salsa20, ERSA, NCS and EHS.**

| Null hypothesis | | $H_0$: All treatment effects are zero | |
|---|---|---|---|
| Alternative hypothesis | | $H_1$: Not all treatment effects are zero | |
| Method | DF | Chi-Square | P-Value |
| Not adjusted for ties | 4 | 22.03 | 0.000 |
| Adjusted for ties | 4 | 22.22 | 0.000 |

P-Value = 0.00 < 0.05(alpha value), which indicates that the difference in the encryption times is statistically significant for the different algorithms and data sizes.

A Post Hoc Bonferroni pairwise comparison test was used to see if there was a significant difference between pairs of algorithms and data sizes. The encryption timings for EHS and NCS were statistically different from Salsa20, Chacha20, and ERSA with P-Values less than 0.05 ($P - Value < 0.05$, $Reject\ H_0$) according to Table 8.

From Table 10, the decryption Times for Salsa20—Chacha20, ERSA -EHS, and NCS -EHS are statistically not different.

## 6.0 Discussions

From Table 1, it could be deduced that the encryption times for Salsa20 and Chacha20 were proportional to the data sizes executed which resulted from the addictive, XORing, and constant distance rotation during execution [37, 38]. ERSA encryption times also showed a proportional relationship between data size and encryption time [17]. This made their encryption times predictable, deterministic, and patterned which confirms the work of Masram et al. [39] and [38, 40–45]. The use of longer keys ensures higher security but results in higher CPU utilization when encryption time is dependent on data size (O (N)) [46]. However, the use of smaller keys is the best employed in cloud computing due to less CPU engagement [47].

The encryption time for NCS and EHS is non-patterned, non-deterministic, and unpredictable because of the disintegration of the keys through the application of a Fixed Sliding Window Algorithm and XORing of the keys and the plaintext which makes NCS and EHS resistant to breaking the resultant cipher through XORing any captured encoded text [48]. Again the randomization from the application of the Sliding Window Algorithm helps to increase the security of the encrypted data and also reduces the time complexity of the Non-Deterministic

**Table 8. Bonferroni simultaneous tests for differences of means of encryption time.**

| Difference of Algorithm Levels | Difference of Means | SE of Difference | Simultaneous 95% CI | T-Value | Adjusted P-Value |
|---|---|---|---|---|---|
| EHS—CHACHA20 | -248.98 | 7.83 | (-271.08, -226.89) | -31.80 | 0.000 |
| ERSA—CHACHA20 | -167.97 | 7.83 | (-190.07, -145.88) | -21.45 | 0.000 |
| NCS—CHACHA20 | -257.23 | 7.83 | (-279.32, -235.13) | -32.85 | 0.000 |
| SALSA20—CHACHA20 | 0.54 | 7.83 | (-21.55, 22.64) | 0.07 | 1.000 |
| ERSA—EHS | 81.01 | 7.83 | (58.92, 103.11) | 10.35 | 0.000 |
| NCS—EHS | -8.24 | 7.83 | (-30.34, 13.85) | -1.05 | 1.000 |
| SALSA20—EHS | 249.53 | 7.83 | (227.43, 271.62) | 31.87 | 0.000 |
| NCS—ERSA | -89.26 | 7.83 | (-111.35, -67.16) | -11.40 | 0.000 |
| SALSA20—ERSA | 168.52 | 7.83 | (146.42, 190.61) | 21.52 | 0.000 |
| SALSA20—NCS | 257.77 | 7.83 | (235.68, 279.87) | 32.92 | 0.000 |

Individual confidence level = 99.50%

Cryptographic Scheme [49, 50]. From the trend of the encryption time from Table 1, it could be concluded that encryption times for NCS and EHS are not dependent on data size but the size of the key while the encryption times for Salsa20, Chacha20, and ERSA are influenced by data size.

From these discussions, it could be concluded that data size is proportional to the decryption time for Salsa20, Chaca20, and ERSA as indicated in Table 2. This makes their trend of decryption time deterministic, predictable, and linear which is supported by the works of [44, 51]. With a linear trend of decryption time, hackers can predict, intercept and modify data [52].

Based on these discussions, it is possible to conclude that ERSA, Salsa20, and Chacha20 produced linear, predictable, deterministic, and high decryption times, making them vulnerable to side-channel attacks and thus do not guarantee the absolute privacy and confidentiality of data on the cloud, as suggested by Kumar et al. [34] and Karthik [35]. The application of the Fixed Sliding Window Algorithm, which disintegrates the huge numbers obtained from the selection of the good prime numbers as the initial keys, Linear Congruential Generator, and XORing the keys and the Ciphertext to obtain plaintext, caused NCS and EHS to have unpredictable, non-deterministic, and non-linear decryption time.

This has the advantage of reducing bandwidth utilization since data encryption and decryption raise the overhead cost of data processing [53]. Again, non-linear encryption timings aided in increasing data secrecy and privacy while reducing device ripping and wear for industry participants and people [18].

According to Tables 1 and 3, encryption time is inversely related to throughput time. Algorithms with faster throughput times use less CPU, and vice versa [54].

When Tables 2 and 4 were compared, it was possible to deduce that with a long decryption time, the associated throughput time was short. This supports the findings of Abolade et al. [54], who discovered that algorithms with a high throughput time need less CPU time.

Table 5 shows that algorithms that consume less memory during execution serve to decrease computational bottlenecks for the CPU and, as a consequence, are regarded as the best [55]. According to Table 6, Salsa20 is more memory intensive since its operation is based on 20 cycles with 10 repeating instances [56]. Because the secret key and Ciphertext are XORed without padding, the NCS had the lowest mean memory usage.

The Friedman Test and Bonferroni Post Hoc test results from Tables 7–10 show that the encryption and decryption times for NCS and EHS are statistically different from Salsa20, Chacha20, and ERSA.

It could be summarized that NCS and EHS produced lower, non-deterministic, non-patterned, and secret key-dependent run times as such defeats the idea behind ERSA, Salsa20, and Chacha20 as the fastest symmetric algorithms which used less memory during data execution [49]. This makes NCS a lightweight algorithm to be employed in the cloud and other areas where fast and lightweight algorithms are needed. Also, it could be used in environments where mobile devices and other devices with less memory are used such as the Internet of Things.

**Table 9. Friedman test of the decryption times for Chacha20, Salsa20, ERSA, NCS and EHS.**

| Null hypothesis | | $H_0$: All treatment effects are zero |
|---|---|---|
| Alternative hypothesis | | $H_1$: Not all treatment effects are zero |
| DF | Chi-Square | P-Value |
| 4 | 22.93 | 0.000 |

**Table 10. Bonferroni simultaneous tests for differences of means for decryption time.**

| Difference of Algorithm Levels | Difference of Means | SE of Difference | Simultaneous 95% CI | T-Value | Adjusted P-Value |
|---|---|---|---|---|---|
| EHS—CHACHA20 | -2000.8 | 32.8 | (-2093.3, -1908.3) | -61.01 | 0.000 |
| ERSA—CHACHA20 | -1997.6 | 32.8 | (-2090.1, -1905.0) | -60.92 | 0.000 |
| NCS—CHACHA20 | -2001.9 | 32.8 | (-2094.5, -1909.4) | -61.05 | 0.000 |
| SALSA20—CHACHA20 | 4.8 | 32.8 | (-87.7, 97.4) | 0.15 | 1.000 |
| ERSA—EHS | 3.2 | 32.8 | (-89.3, 95.7) | 0.10 | 1.000 |
| NCS—EHS | -1.1 | 32.8 | (-93.7, 91.4) | -0.03 | 1.000 |
| SALSA20—EHS | 2005.6 | 32.8 | (1913.1, 2098.2) | 61.16 | 0.000 |
| NCS—ERSA | -4.3 | 32.8 | (-96.9, 88.2) | -0.13 | 1.000 |
| SALSA20—ERSA | 2002.4 | 32.8 | (1909.9, 2095.0) | 61.06 | 0.000 |
| SALSA20—NCS | 2006.8 | 32.8 | (1914.2, 2099.3) | 61.20 | 0.000 |

Individual confidence level = 99.50%

## 7.0 Conclusion

To ensure the secrecy and privacy of data in the cloud, modern cryptographic techniques are applied to encode and decode data. These encryption techniques have computational overheads that have an impact on cloud performance. The symmetric stream cipher algorithms; ERSA, Salsa20, Chacha20, NCS, and EHS have all been thoroughly examined. For securing the privacy and secrecy of data stored in the cloud, ERSA, Salsa20, and Chacha20 are seen to be strong cryptographic schemes. However, compared to NCS and EHS, their run times are linear, predictable, and long, rendering them vulnerable to side-channel attacks.

Their linear runtime trends result in significant bandwidth use and hardware device wear and tear during the transfer of large amounts of data, making them unsuitable for a cloud computing environment. Additionally, because of their linear run times, hackers can estimate the execution time of any piece of data. The Friedman test and Bonferroni Post Hoc test, however, showed that NCS and EHS had the advantage of producing non-linear run time trends, non-patterned run time trends, non-deterministic run time trends, lowest run times, high throughput, and consumed less amount of memory during execution.

Since NCS and EHS will ensure reduced bandwidth usage, prevent tearing and wearing of hardware, and maximize the utilization of any device without much attention on the specification of hardware, this provides industry players and academia optimism that they can fully embrace cloud computing. Future studies should focus on doing experiments using computers with greater specifications. Additionally, research should be done to compare the security strength of NCS to other cutting-edge algorithms.

## Author Contributions

**Conceptualization:** John Kwao Dawson.

**Data curation:** John Kwao Dawson.

**Formal analysis:** John Kwao Dawson.

**Investigation:** John Kwao Dawson.

**Methodology:** John Kwao Dawson.

**Supervision:** Twum Frimpong, James Benjamin Hayfron Acquah, Yaw Marfo Missah.

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
