## [Decision Letter · Decision Letter 0]

8 Jun 2023

PONE-D-23-09341ENSURING PRIVACY AND CONFIDENTIALITY OF CLOUD DATA: A COMPARATIVE ANALYSIS OF DIVERSE CRYPTOGRAPHIC SOLUTIONS BASED ON RUN TIME TRENDPLOS ONE

Dear Dr. Dawson,

Thank you for submitting your manuscript to PLOS ONE. After careful consideration, we feel that it has merit but does not fully meet PLOS ONE’s publication criteria as it currently stands. Therefore, we invite you to submit a revised version of the manuscript that addresses the points raised during the review process.

We look forward to receiving your revised manuscript.

Kind regards,

Pandi Vijayakumar, Ph.D

Academic Editor

PLOS ONE

Journal Requirements:

"No funding"

"No competing interest"

6. We note that Figures 1 and 2 in your submission contain copyrighted images. All PLOS content is published under the Creative Commons Attribution License (CC BY 4.0), which means that the manuscript, images, and Supporting Information files will be freely available online, and any third party is permitted to access, download, copy, distribute, and use these materials in any way, even commercially, with proper attribution. For more information, see our copyright guidelines: http://journals.plos.org/plosone/s/licenses-and-copyright.

a. You may seek permission from the original copyright holder of Figures 1 and 2 to publish the content specifically under the CC BY 4.0 license. 

Additional Editor Comments:

Based on the comments of the reviewers, I recommend this paper for major revision.

Reviewers' comments:

Reviewer's Responses to Questions

**Comments to the Author**

1. Is the manuscript technically sound, and do the data support the conclusions?

Reviewer #1: Yes

Reviewer #2: Partly

2. Has the statistical analysis been performed appropriately and rigorously? 

Reviewer #1: N/A

Reviewer #2: No

3. Have the authors made all data underlying the findings in their manuscript fully available?

Reviewer #1: Yes

Reviewer #2: No

4. Is the manuscript presented in an intelligible fashion and written in standard English?

Reviewer #1: Yes

Reviewer #2: No

5. Review Comments to the Author

Reviewer #1: 1. Two algorithms are defined for non-linear and non-deterministic execution time and five algorithms are defined for produced linear, predictable and deterministic run time. The authors have to identify some other recent algorithms for non-linearity and that must also be included for comparative analysis.

2. The authors must also provide concluding remarks on such comparative analysis in the abstract section itself.

3. Do all the algorithms mentioned in the manuscript ensure both privacy and confidentiality of cloud data?

4. What is the benchmark dataset used in cloud environment to ensure privacy and confidentiality in this manuscript?

5. The manuscript is too lengthy. Some of the unimportant text can be removed.

6. Even though this manuscript concentrates on comparative analysis, some kind of novelty must be provided in a separate section as a base for a new proposed system in relation to the existing algorithms.

7. The performance section can be still improved with advanced parameter comparisons.

8. The related section can also be strengthened with reference to privacy and confidentiality parameters in the existing literature.

Reviewer #2: 1. Authors should assign chapter numbers carefully. There are several mistakes in the numbering throughout the paper. For example, please check the conclusion section number, as well as section 3.4.4, which is mistakenly labeled as 3.3.5.

2. The overall organization of the article is difficult to understand, and it lacks continuity. The authors should revise the structure and flow of the paper to ensure a logical and coherent progression of ideas.

3. Please check equation numbers carefully. There is an instance where equation number 13 is represented twice. Ensure that all equations are properly numbered and referenced.

4. The flow of the article is lacking in explanations about tables and figures. Make sure to provide clear and sufficient explanations for each table and figure to help readers understand their relevance and implications.

5. The manuscript contains numerous grammatical and typographical errors. The authors are advised to thoroughly proofread the paper and consider professional English proofreading services to improve the language quality.

6. The novelty of the work needs to be more explicitly described. The authors should clearly state the unique contributions of their research and highlight how it differentiates from previous works in the field.

6. PLOS authors have the option to publish the peer review history of their article (what does this mean?). If published, this will include your full peer review and any attached files.

Reviewer #1: No

Reviewer #2: No

---

## [Author Response · Author response to Decision Letter 0]

24 Jul 2023

Figure one is the authors own diagram while Figure 2 has been deleted

---

## [Decision Letter · Decision Letter 1]

16 Aug 2023

ENSURING PRIVACY AND CONFIDENTIALITY OF CLOUD DATA: A COMPARATIVE ANALYSIS OF DIVERSE CRYPTOGRAPHIC SOLUTIONS BASED ON RUN TIME TREND

PONE-D-23-09341R1

Dear Dr. Dawson,

We’re pleased to inform you that your manuscript has been judged scientifically suitable for publication and will be formally accepted for publication once it meets all outstanding technical requirements.

Kind regards,

Pandi Vijayakumar, Ph.D

Academic Editor

PLOS ONE

Additional Editor Comments (optional):

Reviewers' comments:

Reviewer's Responses to Questions

**Comments to the Author**

1. If the authors have adequately addressed your comments raised in a previous round of review and you feel that this manuscript is now acceptable for publication, you may indicate that here to bypass the “Comments to the Author” section, enter your conflict of interest statement in the “Confidential to Editor” section, and submit your "Accept" recommendation.

Reviewer #1: All comments have been addressed

Reviewer #2: All comments have been addressed

2. Is the manuscript technically sound, and do the data support the conclusions?

Reviewer #1: Yes

Reviewer #2: Partly

3. Has the statistical analysis been performed appropriately and rigorously? 

Reviewer #1: N/A

Reviewer #2: No

4. Have the authors made all data underlying the findings in their manuscript fully available?

Reviewer #1: Yes

Reviewer #2: No

5. Is the manuscript presented in an intelligible fashion and written in standard English?

Reviewer #1: Yes

Reviewer #2: No

6. Review Comments to the Author

Reviewer #1: (No Response)

Reviewer #2: We have carefully reviewed your revised manuscript, and we appreciate the efforts you have made to address the comments and improve the clarity of the methodology. The revised version provides a more detailed explanation of the cryptographic schemes and their integration. However, there are still some areas that require further clarification and improvement. We kindly request you to consider the following points for the final version of the manuscript:

1. Enhanced RSA (ERSA): The description of Enhanced RSA in the methodology section is not entirely clear. The integration of Gaussian interpolation with the traditional RSA needs further elaboration. Please provide a step-by-step explanation of how the interpolation process enhances the security of RSA, and how it helps overcome the factorization problem.

2. Non-Deterministic Cryptographic Scheme (NCS): While the methodology briefly mentions the use of Good Prime Numbers, Linear Congruential Generator, and Fixed Sliding Window Algorithm, the specific details of how these components are integrated into the NCS are missing. Please provide more comprehensive information on how each component contributes to the security and effectiveness of the NCS.

3. Enhanced Homomorphism Scheme (EHS): Similar to the NCS, the EHS section lacks specific details regarding the integration of Good Prime Numbers, Linear Congruential Generator, Fixed Sliding Window Algorithm, and Gentry's Algorithm. Please provide a clear and step-by-step explanation of how these components are combined to create the EHS.

4. Salsa20 and Chacha20: The methodology provides a brief description of Salsa20 and Chacha20. However, to enhance understanding, please provide a more detailed explanation of the internal workings of these stream ciphers, including the role of quarterround operations and the number of rounds used in each algorithm. Additionally, clarify the specific benefits of Chacha20 over Salsa20, as mentioned in the section.

5. Proposed Framework: In the section discussing the proposed framework, it would be beneficial to provide more information about the key generation, encryption, and decryption phases for each cryptographic scheme (ERSA, NCS, EHS, Chacha20, and Salsa20). Please elaborate on the processes involved in each phase and how they contribute to the overall security and efficiency of the system.

6. Benchmarking and Evaluation: While the methodology mentions that the architecture is divided into five phases for benchmarking (key generation, encryption, decryption, memory utilization, and throughput), specific details on the benchmarking process are missing. Please provide a clear explanation of the evaluation metrics used, the datasets utilized for testing, and the criteria for comparing the different cryptographic schemes.

7. Experiment Reproducibility: It is essential to ensure the reproducibility of the experiments for validation purposes. Please provide detailed information on the hardware and software setup used in the experiments, including the specification of the digital devices and any libraries or frameworks utilized. Also, consider providing access to the datasets and source code to facilitate independent verification of the results.

8. Comparative Study: The comparative study of the cryptographic schemes (ERSA, NCS, EHS, Chacha20, and Salsa20) could benefit from additional insights. Please include a detailed analysis of the strengths and weaknesses of each scheme, considering factors such as encryption/decryption speed, memory usage, and security features. This will help readers understand the trade-offs between the different schemes and make informed decisions.

9. Figures and Tables: Ensure that all figures and tables are properly labeled and referenced in the text. Additionally, double-check the consistency of the notation used in the figures and the methodology to avoid any confusion.

10. Language and Clarity: Proofread the entire manuscript to improve the language and clarity. Correct any grammatical errors and sentence structures to enhance the readability and coherence of the text.

11. Overall, the methodology section presents an interesting and comprehensive approach to cryptographic schemes. However, addressing the above-mentioned points will further strengthen the clarity and depth of the explanation, making the methodology more robust and accessible to readers.

12. Figure 1, titled "Work flow diagram of enhanced RSA," displays a complex network of arrows representing information flow. However, it is not clear what inputs and outputs are involved in each block of the diagram. Clarification is needed regarding the data or information that enters and exits each block to better understand the processes depicted in the diagram. It would be helpful to rephrase the diagram or provide additional information to clearly indicate the inputs and outputs associated with each block to enhance the overall comprehension of the figure.

7. PLOS authors have the option to publish the peer review history of their article (what does this mean?). If published, this will include your full peer review and any attached files.

Reviewer #1: No

Reviewer #2: **Yes: **Dr.RAJKUMAR.S.C

---

## [Editor Report · Acceptance letter]

21 Aug 2023

PONE-D-23-09341R1 

ENSURING PRIVACY AND CONFIDENTIALITY OF CLOUD DATA: A COMPARATIVE ANALYSIS OF DIVERSE CRYPTOGRAPHIC SOLUTIONS BASED ON RUN TIME TREND 

Dear Dr. Dawson:

I'm pleased to inform you that your manuscript has been deemed suitable for publication in PLOS ONE. Congratulations! Your manuscript is now with our production department. 

Kind regards, 

on behalf of

Dr. Pandi Vijayakumar 

Academic Editor

PLOS ONE